# From Systemic Inflammation to Vascular Remodeling: Investigating Carotid IMT in COVID-19 Survivors

**DOI:** 10.3390/v17091196

**Published:** 2025-08-30

**Authors:** Emilia Bielecka, Piotr Sielatycki, Paulina Pietraszko, Sara Anna Frankowska, Edyta Zbroch

**Affiliations:** Department of Internal Medicine and Hypertension, Medical University of Bialystok, 15-089 Bialystok, Poland; piotr.sielatycki@umb.edu.pl (P.S.); pietraszko270@gmail.com (P.P.); sarafrankowsk@gmail.com (S.A.F.); edyta.zbroch@umb.edu.pl (E.Z.)

**Keywords:** atherosclerosis, COVID-19, SARS-CoV-2, ultrasonography, common carotid artery, intima media thickness, IMT

## Abstract

Background: Atherosclerosis is a chronic inflammatory condition that underlies both cardiovascular and cerebrovascular complications. Emerging evidence suggests that COVID-19 may play a role in its progression. Purpose: The aim of the study was to evaluate the potential impact of SARS-CoV-2 infection on the development of atherosclerosis. Patients and Methods: Common carotid artery (CCA) intima media thickness (IMT) was measured by ultrasonography twice, 12–18 months apart, in a cohort of 92 patients (47 with COVID-19 and 45 controls). Clinical data were collected from medical histories, physical examinations, and laboratory findings. Results: Baseline IMT values were comparable between the study groups (0.85 mm vs. 0.78 mm). However, the COVID-19 group exhibited a significantly greater increase in IMT over time, with a median change of 0.13 mm compared to 0.05 mm in the controls (*p* = 0.018). Furthermore, 69.2% of COVID-19 patients exceeded the median IMT progression threshold compared to 36% in the control group (*p* = 0.017). An elevated level of C-reactive protein (CRP) and a higher triglyceride (Tg)-to-High-Density Lipoprotein Cholesterol (HDL) ratio were significantly associated with increased IMT in the COVID-19 group. Age and heart rate were identified as significant predictors of IMT progression across both groups. Conclusions: COVID-19 may accelerate the progression of subclinical atherosclerosis. The strong associations of CRP and the TG/HDL ratio with IMT highlight the potential roles of chronic inflammation and metabolic dysregulation in driving these vascular changes. Further large-scale, multicenter studies are needed to elucidate the underlying mechanisms, confirm these observations, and guide targeted preventive and therapeutic strategies for individuals with an increased cardiovascular and cerebrovascular risk.

## 1. Background

Atherosclerosis (AS) is a chronic inflammatory disease of the arteries characterized by the accumulation of lipids, immune cells, and fibrous tissue within the vascular wall, leading to progressive narrowing of the arterial lumen. This process significantly increases the risk of cardiovascular and cerebrovascular complications, which are among the leading and increasingly prevalent causes of morbidity and mortality worldwide [1,2,3,4,5].

The development of atherosclerosis is a multifactorial and complex process involving numerous pathophysiological mechanisms [6,7]. Well-established risk factors include hypercholesterolemia, hypertension, diabetes mellitus, and smoking. Despite efforts to address these traditional risk factors, the global burden of atherosclerotic cardiovascular disease (ASCVD) remains high [8,9,10]. Furthermore, atherosclerosis is frequently diagnosed in individuals without recognized risk factors [11], prompting researchers to investigate novel contributing mechanisms. One such area of interest is the role of inflammation in the pathogenesis of atherosclerosis [12].

Recent studies have also suggested associations between subclinical atherosclerosis and chronic infections caused by viruses such as hepatitis C virus (HCV), human immunodeficiency virus (HIV), influenza virus, cytomegalovirus (CMV), Epstein–Barr virus (EBV), herpes simplex virus types 1 and 2 (HSV-1, HSV-2), human T cell leukemia virus type 1 (HTLV-1), and even Mycobacterium tuberculosis [13,14,15,16].

Preclinical atherosclerosis refers to early arterial wall changes that may be present before the onset of clinical symptoms [17]. Carotid intima–media thickness (IMT) is widely accepted as a non-invasive marker of subclinical atherosclerosis and a strong predictor of future cardiovascular and cerebrovascular events [18]. IMT is measured using carotid ultrasound and enables the identification of individuals who may benefit from early cardiovascular risk assessments and interventions. Moreover, serial IMT assessments are valuable for monitoring disease progression and evaluating therapeutic efficacy [19,20].

Severe acute respiratory syndrome coronavirus 2 (SARS-CoV-2), the virus responsible for COVID-19, emerged in December 2019 in Wuhan, China, and rapidly overwhelmed healthcare systems worldwide [21]. It is now evident that SARS-CoV-2 infection can lead to persistent, multisystem complications, collectively referred to as long COVID [22,23,24,25,26,27,28,29,30,31,32,33,34]. Millions of individuals worldwide are affected, and the number continues to grow [35]. Although the long-term trajectory of post-COVID complications remains unclear, emerging data suggest a sustained increase in the risk of ASCVD, persisting up to one-year post-infection [36,37,38].

Some studies propose that SARS-CoV-2 may contribute to atherosclerosis development through mechanisms including systemic inflammation, chronic endothelial dysfunction, disruption of the endothelial barrier, increased oxidative stress, and a prothrombotic state [39,40,41,42,43,44,45,46,47]. While a direct causal relationship between COVID-19 and atherosclerosis has not been definitively established, it is plausible that the known consequences of SARS-CoV-2 infection may exacerbate traditional risk factors for atherosclerosis [48,49,50].

In addition, lifestyle changes during the pandemic—such as prolonged isolation, increased psychological stress, and reduced physical activity—have led to greater incidences of sedentary behavior, weight gain, and obesity, all of which are independently associated with atherosclerosis risks [51,52].

Our study addresses one of the major challenges in contemporary medicine: the early detection and prevention of atherosclerosis. It is imperative to determine whether SARS-CoV-2 infection influences carotid IMT, as this could have significant implications for cardiovascular risk stratification and management. To this end, we evaluated changes in IMT using Doppler-enhanced ultrasonography.

The aim of the study was to determine the potential impact of SARS-CoV-2 infection on the development of subclinical atherosclerosis, as assessed by changes in common carotid artery intima–media thickness.

## 2. Materials and Methods

### 2.1. Study Design and Participants

We present a single-center, prospective cohort study conducted at the Department of Internal Medicine and Hypertension, Medical University of Bialystok, Poland. Patient recruitment was carried out between 26 October 2021 and 30 September 2024.

Participants were consecutively enrolled among individuals hospitalized for various medical conditions during the study period.

Participants eligible for inclusion were adults over 18 years of age with no prior diagnosis of carotid atherosclerosis who provided written informed consent. Exclusion criteria included pregnancy or breastfeeding, a previous diagnosis of carotid atherosclerosis, or failure to provide informed consent.

The study was approved by the Bioethics Committee of the Medical University of Bialystok (Resolution No. APK.002.256.2022).

A total of 92 participants were enrolled and stratified into two groups based on SARS-CoV-2 infection history. The COVID-19 group included individuals with a confirmed current or past SARS-CoV-2 infection, verified via medical records, patient history, or laboratory test results. The control group comprised participants with no reported history of COVID-19.

Within the COVID-19 group, disease severity was categorized based on the clinical course as follows:

Mild: Patients who experienced a mild, ambulatory illness without requiring hospitalization.

Moderate/Severe: Patients who required hospitalization due to COVID-19, indicating a more severe disease course.

### 2.2. Clinical Data Collection

Clinical and laboratory data were collected retrospectively from electronic medical records. The extracted variables included demographic information (age and sex), vital signs on admission (heart rate and respiratory rate), and laboratory parameters such as C-reactive protein (CRP), creatinine, total cholesterol, Triglycerides (TG), high-density lipoprotein cholesterol (HDL), and low-density lipoprotein cholesterol (LDL). No additional blood samples were obtained for the purpose of this study. To enhance the assessment of lipid metabolism, lipid ratios were calculated, including TG/HDL, total cholesterol/HDL, and LDL/HDL. A detailed review of each patient’s medical history was also performed. This included smoking status, the presence of comorbidities (e.g., hypertension, diabetes mellitus, heart failure, and prior thrombotic events such as myocardial infarction or stroke), and current pharmacological treatments. Medications assessed comprised statins, angiotensin-converting enzyme (ACE) inhibitors, angiotensin receptor blockers (ARBs), beta-blockers, digitalis glycosides, calcium channel blockers, alpha-blockers, diuretics, aldosterone antagonists, anticoagulants, hypoglycemic agents, sedatives, and acetylsalicylic acid. All clinical and historical data were collected at the time of the first intima–media thickness (IMT) assessment.

### 2.3. Intima–Media Thickness (IMT) Assessment

IMT measurements were performed in accordance with the guidelines of the Polish Society of Vascular Surgery for duplex Doppler ultrasound examination of the carotid and vertebral arteries. A Philips EPIQ 7 Elite ultrasound system equipped with a Philips L12-3 linear transducer operating at 12–13 MHz was used.

IMT was assessed in the distal segment of the common carotid artery (CCA), approximately 10 mm proximal to the carotid bulb, using longitudinal views. High-resolution images were obtained for both left and right CCAs, and the final IMT value was calculated as the average of three measurements on each side.

All assessments were performed by a single experienced sonographer, who completed specific training in vascular ultrasound to ensure methodological consistency and minimize interobserver variability.

IMT measurements were performed twice, with an interval of 12 to 18 months between the first and follow-up examination, to assess the progression of subclinical atherosclerosis.

### 2.4. Follow-Up

Follow-up data on intima–media thickness (IMT) were successfully obtained for 26 participants in the COVID-19 group and for 25 participants in the control group.

Reduced follow-up participation was primarily due to logistical challenges, including transportation difficulties. Many participants resided in remote areas without access to private vehicles or hospital-arranged transport.

### 2.5. Statistical Analysis

All statistical analyses adhered to a significance level of α = 0.05.

The distribution of numerical variables was assessed using non-parametric methods to account for the lack of normality in clinical parameters. These variables were summarized as medians with interquartile ranges (IQR). Categorical variables were described as absolute counts and percentages within specific groups. Statistical differences in numerical variables between two independent groups were determined using the Wilcoxon rank-sum test, while the Kruskal–Wallis test was applied to assess differences among three or more independent groups. Independence between categorical variables was evaluated using Pearson’s Chi-square test; when expected frequencies were low, Fisher’s exact test was utilized.

The relationships between two continuous variables were explored using Spearman’s rank correlation coefficient (Rho). For these analyses, the confidence interval (95% CI) at 95% and *p*-values were calculated using asymptotic approximations of the *t*-test.

To examine the adjusted effects of predictors while accounting for potential confounders, multiple linear regression models with robust estimators were employed. These models estimated the association between predictor variables and continuous outcomes, providing 95% CIs and *p*-values derived from the *t*-test. For dichotomous outcomes, a multiple logistic regression with a logit link function was used to estimate the adjusted effects of predictors. The significance of these associations was assessed using Wald z-statistics, with 95% CIs and *p*-values reported.

No formal correction for multiple comparisons was applied. This decision is based on the exploratory nature of the study and the relatively small sample size, where conservative adjustments such as Bonferroni or FDR could substantially increase the risk of a Type II error and obscure potentially meaningful associations.

Analyses were conducted using the R Statistical language (version 4.3.3) on Windows 11 pro 64 bit, using the packages sjPlot (version 2.8.15), parameters (version 0.22.2), performance (version 0.12.3), report (version 0.5.8), correlation (version 0.8.5), gtsummary (version 1.7.2), MASS (version 7.3.60.0.1), and dplyr (version 1.1.4).

## 3. Results

### 3.1. Studied Population

The analyzed group included 92 patients aged 30–92 years, with a female-to-male ratio of 1.56:1. Among them, 47 (51.1%) had a history of COVID-19 while 45 (48.9%) served as the control group. Table 1 presents the clinical characteristics of the studied population.

The median age was 70 years (IQR: 57.75–78.25), with no significant difference between the COVID-19 and control groups (*p* = 0.392). Females constituted 60.87% of the overall sample, with similar distributions in both groups (COVID-19: 63.83%, control: 57.78%, and *p* = 0.552). The age and sex distribution was comparable. Heart rate was significantly higher in the COVID-19 group (73 bpm vs. 70 bpm, *p* = 0.047).

CRP levels were significantly elevated in the COVID-19 group (8.45 mg/L vs. 1.08 mg/L, *p* = 0.003). The triglyceride-to-HDL ratio was also higher (2.59 vs. 1.82, *p* = 0.045). Medication use, including statins, antihypertensives, and anticoagulants, was similar across groups, as shown Table 2.

### 3.2. IMT Measurements

Among COVID-19 patients, 66% were in the acute stage at the time of examination, and 59.6% had moderate to severe disease; these proportions are shown in Figure 1.

At baseline, the median IMT was 0.85 mm in the COVID-19 group and 0.78 mm in the control group. IMT on the first and second measurement in every sample is shown in Figure 2.

The COVID-19 group exhibited a greater IMT increase (1.00 mm vs. 0.86 mm, *p* = 0.19). Notably, 69.2% of COVID-19 patients had an IMT increase above the median value (0.08 mm), compared to only 36% of the control group (*p* = 0.017), which is illustrated in Figure 3.

The difference in IMT (Δt2 − t1) was significantly greater in the COVID-19 group (median: 0.13 mm) than in the control group (median: 0.05 mm, *p* = 0.018). The variation is shown in Figure 4.

Figure 5 displays IMT differences at the first measurement and its progression across disease phases in the COVID-19 group.

### 3.3. The Relationship Between IMT at the First Measurement and Clinical and Demographic Parameters

Table 3 highlights factors associated with the baseline IMT. Age emerged as the strongest predictor across all groups, showing a significant positive correlation in the overall sample (Rho = 0.53, *p* < 0.001), as well as within the COVID-19 (Rho = 0.49, *p* < 0.001) and control groups (Rho = 0.58, *p* < 0.001).

Among metabolic markers, the triglyceride-to-HDL (TG/HDL) ratio showed a significant positive correlation with IMT in the overall sample (Rho = 0.22, *p* = 0.038), with a stronger correlation in the COVID-19 group (Rho = 0.34, *p* = 0.022).

In the COVID-19 group, lipid profiles correlated with IMT and HDL cholesterol showed a negative association (Rho = −0.33, *p* = 0.029); LDL cholesterol (Rho = −0.31, *p* = 0.039) and total cholesterol (Rho = −0.30, *p* = 0.044) were also inversely related to IMT.

Additionally, in the control group, a longer time between measurements was inversely correlated with the baseline IMT (Rho = −0.43, *p* = 0.031).

### 3.4. Association Between IMT Changes Across Measurements and Clinical and Demographic Parameters

The change in IMT between the second and first measurements reported in Table 4 was evaluated for associations with various clinical and demographic parameters. In the overall sample, heart rate (HR) showed a significant positive correlation with IMT progression (Rho = 0.35, 95% CI: 0.07 –0.57, and *p* = 0.013).

HDL cholesterol levels demonstrated a significant negative correlation with IMT progression in the overall sample (Rho = −0.32, 95% CI: −0.56–−0.04, and *p* = 0.022).

In the COVID-19 group, heart rate was also significantly correlated with IMT progression (Rho = 0.39, 95% CI: −0.01–0.68, and *p* = 0.049). Additionally, the HDL cholesterol level was inversely associated with the change in IMT (Rho = -0.43, 95% CI: −0.71–−0.03, and *p* = 0.032).

In the control group, no parameter was significantly associated with IMT progression.

Table 5 and Table 6 present the differences in intima–media thickness (IMT) values at the first measurement across groups defined by selected categorical parameters. In both tables, the analysis was conducted for the overall sample and separately within the COVID-19 and control groups. For each comparison, Group 1 refers to individuals presenting a given characteristic (e.g., smokers, patients with hypertension, and individuals taking a specific medication), whereas Group 2 comprises individuals without that characteristic (e.g., non-smokers, patients without hypertension, or those not receiving the respective drug). Each table displays the median IMT values (with interquartile ranges) for both groups, along with *p*-values assessing the statistical significance of the differences observed.

A significant difference was observed in IMT changes between the COVID-19 and control groups. The COVID-19 group exhibited a greater increase in IMT over time (0.13 mm, 95% CI: 0.06–0.50) compared to those without COVID-19 (0.05 mm, 95% CI: 0.01–0.20, and *p* = 0.018).

In the COVID-19 group, changes in IMT were analyzed based on disease stage and severity. While not reaching statistical significance, individuals with acute COVID-19 showed a trend toward greater IMT increases (0.19 mm, 95% CI: 0.06–0.69) compared to those with non-acute disease (0.10 mm, 95% CI: 0.06–0.13, and *p* = 0.106). Similarly, those with moderate to severe COVID-19 tended to exhibit higher IMT progression (0.19 mm, 95% CI: 0.09–0.63) compared to those with mild disease (0.08 mm, 95% CI: 0.05–0.12, and *p* = 0.087).

In the control group, no statistically significant differences were found in IMT changes over time across the analyzed parameters.

### 3.5. Changes in IMT Between Second and First Measurements Based on Baseline IMT Levels

Patients were categorized into three groups based on their initial IMT values: ≤0.6 mm, 0.6–1.0 mm, and >1.0 mm, with comparisons conducted separately for the COVID-19 and control groups.

In the COVID-19 group, IMT progression varied across baseline IMT categories, as shown in Figure 6 and Figure 7.

### 3.6. The Estimation of the Adjusted Effects of Demographic and Clinical Parameters on the Change in IMT

The estimation of the adjusted effects of demographic and clinical parameters on the change in IMT is shown in Table 7 and Table 8.

In the overall cohort, higher HR and a history of vascular incidents were significantly associated with an increase in IMT over time ((β = 5.90 × 10^−3^, 95% CI: 2.50 × 10^−3^–9.30 × 10^−3^, and *p* = 0.001), (β = 0.28, 95% CI: 0.07–0.49, and *p* = 0.011)).

In the COVID-19 group, acute infection at the time of the first measurement was a significant predictor of a smaller change in IMT over time compared to the non-acute group (β = −0.27, 95% CI: −0.54–−1.30 × 10^−3^, and *p* = 0.049).

A trend was observed for greater IMT increases in individuals with moderate to severe COVID-19 courses compared to those with mild disease (β = 0.24, 95% CI: −0.04–0.51, and *p* = 0.092), though this did not reach statistical significance. A higher HR was also associated with IMT progression in the COVID-19 group (β = 0.01, 95% CI: 7.00 × 10^−4^–0.01, and *p* = 0.033).

In the control group, no significant predictors of IMT changes were identified.

## 4. Discussion

Since the onset of the COVID-19 pandemic, caused by the SARS-CoV-2 virus, the disease has been the subject of extensive research, primarily due to its significant impact on human health. Growing evidence points to long-term health consequences, particularly involving the cardiovascular system. Among these potential sequelae are chronic inflammation, metabolic disturbances, and endothelial dysfunction—all of which have been observed in individuals who have recovered from COVID-19. These factors are recognized contributors to the progression of atherosclerosis, a fundamental pathological process underlying cardiovascular disease.

However, the specific mechanisms and risk factors underlying vascular changes in post-COVID-19 patients remain poorly understood. Therefore, the aim of this study was to investigate the potential association between prior COVID-19 infection and subclinical atherosclerosis, assessed through changes in carotid intima–media thickness (IMT).

Our findings demonstrate significant differences in vascular changes between individuals with a history of COVID-19 and those in the control group. The most notable observation was a markedly accelerated increase in IMT among the COVID-19 group, suggesting a more rapid progression of atherosclerosis. Specifically, patients with prior COVID-19 infection exhibited a mean IMT increase of 0.12 mm compared to only 0.04 mm in the control group. Furthermore, 69.2% of patients in the COVID-19 group showed an IMT increase above the median value (0.08 mm), in contrast to 36% in the control group.

Studies have shown that COVID-19 may accelerate atherosclerosis through multiple, interconnected mechanisms. SARS-CoV-2 infects endothelial cells via ACE2, reducing nitric oxide availability, impairing vasodilation, and increasing vascular stiffness [53,54]. The infection triggers a systemic inflammatory response, including elevated IL-6, TNF-α, and IL-1β, which amplifies vascular injury [55]. Dysregulation of the renin–angiotensin–aldosterone system further promotes vasoconstriction, inflammation, and smooth muscle proliferation, while a prothrombotic state increases the risk of plaque formation and instability [53,54].

Persistent vascular damage can remain after viral clearance, driven by low-grade inflammation, extracellular matrix remodeling, endothelial-to-mesenchymal transition, and immune dysregulation, potentially sustained by epigenetic changes [56,57]. In some patients with long COVID, either PCR-negative or trace PCR-positive viral detection has been reported, with rare cases of persistent PCR-positive shedding beyond 3 months after the initial infection [58]. Such persistence may be associated with an expanded and sustained SARS-CoV-2-specific CD8^+^ T cell response, as observed in Danish post-COVID cohorts, which could further contribute to chronic vascular inflammation and remodeling long after the acute infection has resolved. This supports and justifies the validity of our findings [58,59,60].

Moreover, elevated CRP levels in COVID-19 patients support the hypothesis that persistent inflammation plays a role in atherosclerosis progression [61]. Although CRP was measured only during the initial visit, this finding is consistent with prior studies, such as that by Yi-Ping Gao et al. [62], who found persistently elevated hsCRP and TNF-α levels in COVID-19 survivors nearly a year after infection. This reinforces the notion that chronic inflammation may persist long after recovery, continuing to impact vascular health.

Our analysis suggests that age may modulate the extent of IMT changes in the context of SARS-CoV-2 infection, consistent with prior studies showing increased vascular susceptibility in older individuals, because age is a well-known independent risk factor for atherosclerosis and vascular remodeling [63].

In our regression analysis, acute infection at the baseline was associated with a smaller change in IMT over time, which may seem counterintuitive One possible explanation is that, as less time had elapsed since infection, there was less opportunity for vascular remodeling to occur. The effects of COVID-19 on IMT likely develop over a longer period of time, so patients infected earlier show a greater progression. However, this interpretation remains tentative and should be viewed with caution, as it is based on a relatively small subgroup and may be influenced by chance variation. Confirmation with larger longitudinal cohorts will be essential.

Among the lipid parameters analyzed, the TG/HDL ratio showed the strongest association with IMT thickening in the COVID-19 group. This finding suggests that metabolic abnormalities, such as insulin resistance and dyslipidemia, may substantially contribute to atherosclerotic lesion development in this population. Importantly, this ratio may serve as a valuable marker for identifying individuals with an increased cardiovascular risk [64].

Another noteworthy finding was the significant inverse relationship between HDL levels and IMT progression in the COVID-19 group, underscoring the protective role of HDL and the need for lipid profile optimization in post-COVID-19 patients [65]. These results align with emerging evidence suggesting that individuals recovering from COVID-19 may be at an increased risk of early-onset atherosclerosis and underscore the importance of preventive measures [66,67,68,69]. Reports of increased rates of ischemic stroke and myocardial infarction following COVID-19 support this concern [66]. Additionally, genetic studies have revealed overlapping genetic pathways between COVID-19 and atherosclerosis, suggesting potential shared mechanisms underlying both conditions [70,71].

Although limited, the existing literature offers some comparative data. A meta-analysis of vascular function in post-COVID-19 patients demonstrated an impaired brachial artery flow-mediated dilation and altered pulse wave velocity, both of which indicate endothelial dysfunction and subclinical atherosclerosis [72]. These changes are predictive of future ASCVD events [73,74,75]. In a study by Danuta Loboda et al. [76], post-COVID-19 patients demonstrated a significant correlation between arterial stiffness and 10-year atherosclerotic cardiovascular disease (ASCVD) risks based on the SCORE2 algorithm. Similarly, Mario Podrug et al. [77] documented significant vascular changes within three months after infection when compared with pre-pandemic baseline measurements.

To the best of our knowledge, only five studies have investigated IMT in the context of COVID-19 in relation to atherosclerosis and cardiovascular risk. Jud et al. [78] found increased IMT (0.59 (0.52–0.68)) in post-COVID-19 patients compared with controls (0.44 (0.40–0.45)), though lower than in patients with known ASCVD (0.72 (0.60–1.01)). However, their cross-sectional design precluded the longitudinal assessment of IMT changes or the determination of causality. Szeghby et al. [79] reported no significant difference in IMT in young, healthy adults within weeks of infection, although markers of arterial stiffness were elevated. A follow-up study by the same group noted a slight increase in IMT at six months, though not statistically significant [80]. However, the findings are limited by the small sample size, lack of a control group, and absence of serial IMT measurements. Bezerra et al. [81] observed both increases and subsequent decreases in IMT among severely ill COVID-19 patients, suggesting that IMT may fluctuate depending on disease severity and recovery. Chen et al. [82] conducted a longitudinal study and found higher IMT in both asymptomatic/mild and severe/critical COVID-19 groups compared to controls. A modest IMT increase was observed over time in the asymptomatic/mild group, although the reasons remain unclear.

Taken together, only two of these studies showed statistically significant IMT elevations post-COVID-19, which aligns with our findings. Nevertheless, the work of Szeghby et al. [79,80] suggests a possible long-term risk that may manifest later. These findings support the hypothesis that COVID-19 infection may accelerate atherosclerotic changes, even in patients without overt cardiovascular disease. While our results indicate a statistically significant difference in carotid IMT progression between individuals with and without prior COVID-19, this finding alone does not fully establish the primary role of SARS-CoV-2 in accelerating atherosclerosis. Unmeasured factors, such as lipoprotein(a) levels or subtle differences in cardiovascular risk and comorbidities, may have influenced the results [83]. However, our multivariable regression analysis (Table 7) partially addresses these concerns by adjusting for several demographic and clinical parameters. In the overall cohort, a higher heart rate and history of vascular incidents were independently associated with a greater IMT progression, while in the COVID-19 group, disease severity showed a trend toward a greater increase in IMT, and a higher heart rate remained a significant predictor. These results indicate that while traditional risk factors remain important, COVID-19 may have an additional, independent effect. Nevertheless, future studies incorporating a broader panel of vascular and biochemical markers, including lipoprotein(a) and imidazole propionate, will be essential to more precisely quantify the independent impact of SARS-CoV-2 infection [84].

### 4.1. Limitations

The primary limitation of our study is its relatively small sample size, which may have limited the statistical power to detect subtle differences and reduced the generalizability of the findings. The lack of serological testing (e.g., anti-nucleocapsid antibodies) represents an additional limitation, as it may have led to the inclusion of participants with prior asymptomatic SARS-CoV-2 infection in the control group. Given the high local seroprevalence (>90%), such misclassification is likely [85]. However, this would be expected to attenuate, rather than inflate, between-group differences, suggesting that the true effect may be underestimated. This context justifies our decision to define groups based on symptomatic status, while also underscoring the importance of incorporating serological screening in future studies to improve group classification accuracy.

A further limitation is that, in some cases, the initial IMT measurement was conducted during the acute phase of infection. Nevertheless, our comparative analysis between acute-phase and post-recovery measurements did not show significant differences, suggesting only a minimal influence of acute-phase vascular changes.

Moreover, we did not collect detailed information on other important determinants of atherosclerosis progression, such as dietary habits, physical activity levels, and socioeconomic status. While pandemic-related lifestyle changes (e.g., reduced physical activity, altered diet) could have confounded the interpretation of vascular changes, we were unable to adjust for these factors in our analysis. Importantly, the study groups did not differ significantly in key baseline characteristics, including BMI and the prevalence of comorbidities, which suggests broadly similar lifestyle patterns. Nevertheless, unmeasured lifestyle differences remain a potential limitation and should be addressed in future longitudinal studies.

A drawback of our study is the potential for an inflated Type I error due to multiple statistical tests. Because of the exploratory nature of the analyses and the relatively small sample size, we did not apply formal multiple comparison corrections, which would have substantially reduced the statistical power and ability to detect potential associations. Instead, the interpretation of findings was based on effect sizes, confidence intervals, and their consistency with prior evidence, rather than on *p*-values alone. It will be necessary for future studies with larger cohorts to confirm these associations under more stringent statistical adjustments.

One further limitation concerns the relatively high loss to follow-up (45%), which raises the possibility of attrition bias. Although the primary reasons for dropout were logistical, participants who were lost may have differed systematically from those who remained, which could have influenced the observed associations. Therefore, the results should be interpreted with caution, and replication in larger cohorts with better follow-up retention is warranted.

### 4.2. Strengths

Despite its limitations, the study provides novel insights into IMT changes in post-COVID-19 patients, addressing a gap in the current literature. Among studies of this nature, our work stands out due to its relatively large COVID-19 group and matched control cohort.

The longitudinal design—with follow-up IMT measurements taken 12–18 months apart—allowed for an assessment of vascular changes over time. Furthermore, all ultrasound assessments were performed by the same experienced investigator using a standardized protocol recommended by the Polish Society of Vascular Surgery [86]. This consistency enhances the reliability and comparability of the data.

### 4.3. Future Research Directions

To build upon our findings, future studies should include larger cohorts to improve the statistical power and enhance the accuracy of mechanistic insights into COVID-19-associated vascular changes. Prospective, multicenter investigations will be critical in validating our results and in identifying causal links between COVID-19 and atherosclerosis progression.

In particular, a focus on metabolic dysfunction—including insulin resistance, chronic inflammation, and lipid disturbances—may help elucidate the pathophysiological mechanisms involved. While our study focused on IMT, the tunica media-to-intima thickness ratio is also an important early predictor of atherosclerosis [87]. Our imaging protocol did not permit separate measurements, but we note its relevance and recommend it for future research. The use of advanced imaging techniques, endothelial function assessments, and genetic profiling could provide additional layers of understanding.

The results of large-scale projects, such as the CARTESIAN study (“COVID-19 Effects on Arterial Stiffness and Vascular Aging”), expected in 2033, are eagerly anticipated and may shed light on long-term vascular aging and cardiovascular risk following COVID-19 [88].

We hope that the present findings stimulate further inquiry and contribute to the development of effective prevention and treatment strategies for individuals with an elevated cardiovascular risk following SARS-CoV-2 infection.

## 5. Conclusions

Our study suggests that symptomatic COVID-19 infection may accelerate the progression of subclinical atherosclerosis, as indicated by increased carotid IMT. This effect appears to be influenced by metabolic disturbances—particularly elevated TG/HDL ratios—and chronic inflammation, marked by raised CRP levels.

Despite limitations such as a small sample size and lack of variant-specific data, the findings support the need for cardiovascular monitoring in post-COVID-19 patients, especially those with metabolic risk factors. Larger long-term studies are needed to clarify the underlying mechanisms and guide prevention strategies.

## Figures and Tables

**Figure 1 viruses-17-01196-f001:**
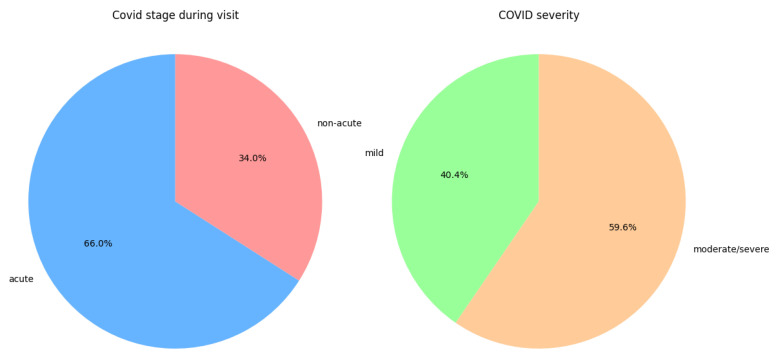
Distribution of disease stage during the first IMT measurement and the distribution of disease severity in the COVID-19 group.

**Figure 2 viruses-17-01196-f002:**
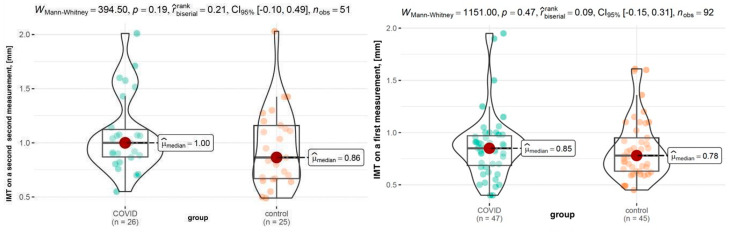
IMT on the first and second measurement.

**Figure 3 viruses-17-01196-f003:**
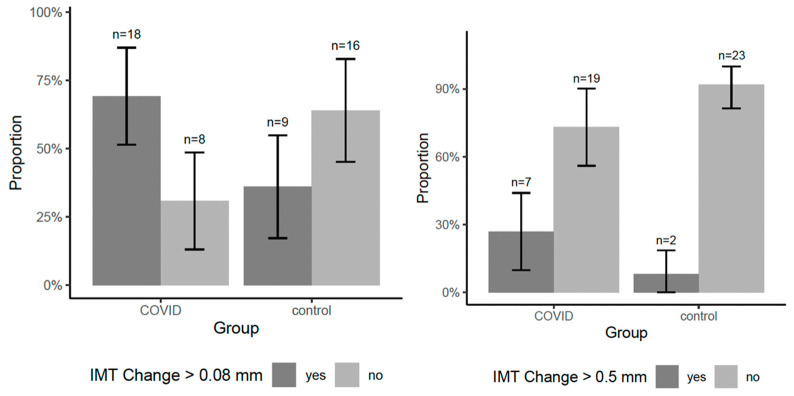
Group-wise proportion of participants with IMT progression exceeding the median and 0.5 mm.

**Figure 4 viruses-17-01196-f004:**
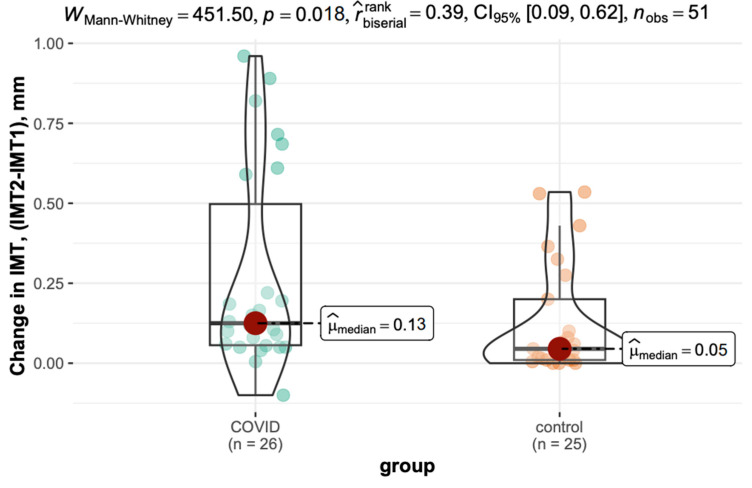
Inter-measurement variation in IMT in the COVID-19 and control groups.

**Figure 5 viruses-17-01196-f005:**
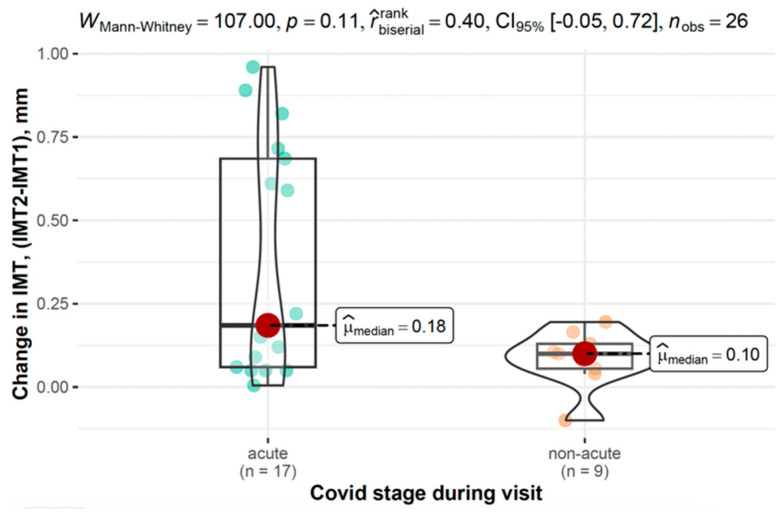
Differences in IMT between the first and second measurements across the disease phase in the COVID-19 group.

**Figure 6 viruses-17-01196-f006:**
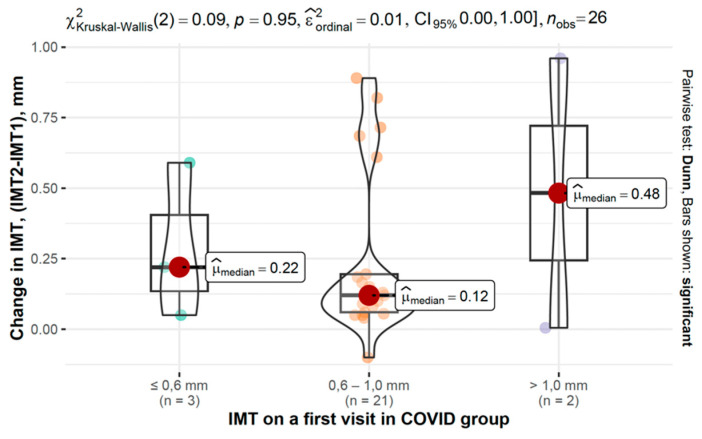
Changes in IMT between second and first measurements based on baseline IMT in the COVID-19 group.

**Figure 7 viruses-17-01196-f007:**
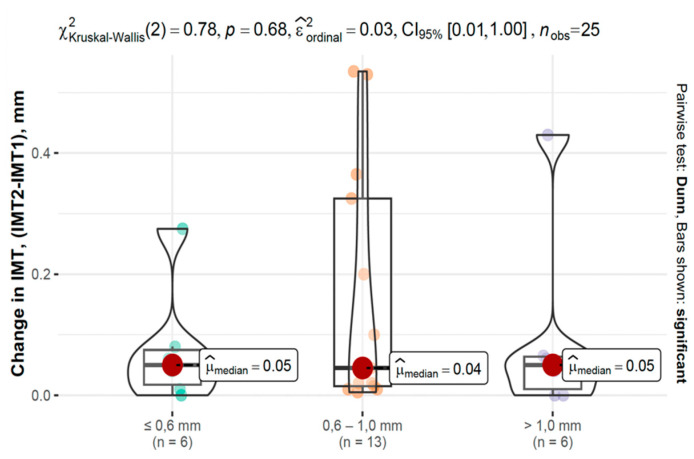
Changes in IMT between second and first measurements based on baseline IMT in the control group.

**Table 1 viruses-17-01196-t001:** Clinical characteristics of the study population.

Characteristic	COVID-19 Group n_1_ = 47 ^a^	Control Group n_2_ = 45 ^a^	*p* ^c^
BMI, kg/m^2^	29.30 (25.93, 31.24) ^b^	28.69 (26.45, 30.63) ^b^	0.761
Smoking status	5 (10.64%)	8 (17.78%)	0.326
SBP, mmHg	130.00 (119.50,145.50) ^b^	129.00 (117.00, 140.00) ^b^	0.628 ^e^
DBP, mmHg	82.00 (73.50, 85.00) ^b^	80.00 (74.00, 85.00) ^b^	0.821 ^e^
HR, bpm	73.00 (69.00, 80.00) ^b^	70.00 (60.00, 76.00) ^b^	**0.047** ^e^
*Comorbidities*			
DM2	11 (23.40%)	9 (20%)	0.692
HT	30 (63.83%)	33 (73.33%)	0.327
Heart failure	21 (44.68%)	13 (28.89%)	0.117
Vascular incident in the past (heart attack, stroke)	11 (23.40%)	4 (8.89%)	0.060 ^d^
Laboratory parameters			
CRP	8.45 (1.53, 37.13) ^b^	1.08 (0.00, 9.00) ^b^	**0.003** ^e^
Creatinine	0.86 (0.69, 0.98) ^b^	0.90 (0.77, 1.05) ^b^	0.271 ^e^
Total Cholesterol [mg/dL]	157.00 (129.00, 189.00) ^b^	165.50 (135.00, 197.25) ^b^	0.311 ^e^
LDL	85.00 (55.00, 108.00) ^b^	86.00 (65.00, 116.25) ^b^	0.275 ^e^
HDL	46.00 (36.00, 55.00) ^b^	50.00 (42.00, 61.00) ^b^	0.100 ^e^
TG	105.00 (80.00, 155.00) ^b^	95.00 (81.75, 129.50) ^b^	0.282 ^e^
TG/HDL	2.59 (1.67, 3.48) ^b^	1.82 (1.27, 2.84) ^b^	**0.045** ^e^
CHOL/HDL	3.34 (2.73, 4.32) ^b^	2.94 (2.52, 3.80) ^b^	0.208 ^e^
LDL/HDL	1.67 (1.35, 2.44) ^b^	1.57 (1.22, 2.04) ^b^	0.568 ^e^

Note: Bold—statistically significant, ^a^ n (%),^b^ median (IQR), ^c^ Pearson’s Chi-squared test, ^d^ Fisher’s exact test, and ^e^ Wilcoxon rank-sum test. BMI—Body Mass Index, SBP—Systolic Blood Pressure, DBP—Diastolic Blood Pressure, HR—heart rate, DM2—Diabetes Mellitus type 2, HT—hypertension, CRP—C-Reactive Protein, LDL—Low-Density Lipoprotein Cholesterol, HDL—High-Density Lipoprotein Cholesterol, and TG—Triglycerides.

**Table 2 viruses-17-01196-t002:** Medication use in the study population.

Medication Intake	COVID-19 Group n_1_ = 47 ^a^	Control Group n_2_ = 45 ^a^	*p* ^b^
Statin	25 (53.19%)	22 (48.89%)	0.680
ACEI	20 (42.55%)	21 (46.67%)	0.692
ARB	7 (14.89%)	9 (20%)	0.518
Beta-blocker	28 (59.57%)	20 (44.44%)	0.146
Digoxin	0 (0%)	2 (4.55%)	0.231 ^c^
Calcium Channel Blocker	13 (27.66%)	20 (44.44%)	0.093
Alpha-blocker	3 (6.38%)	6 (13.33%)	0.311 ^c^
Diuretic	20 (42.55%)	20 (44.44%)	0.855
Aldosterone Antagonist	5 (10.64%)	10 (22.22%)	0.133
Sedatives/Hypnotics	11 (23.40%)	13 (28.89%)	0.549
Anticoagulants	14 (29.79%)	10 (22.22%)	0.409
Hypoglycemic Agents	11 (23.40%)	9 (20%)	0.692
Acetylsalicylic Acid	9 (19.15%)	7 (15.56%)	0.649

Note: ^a^ n (%), ^b^ Pearson’s Chi-squared test, ^c^ Fisher’s exact test, and ACEI—Angiotensin-Converting Enzyme Inhibitors, ARB—Angiotensin II Receptor Blockers.

**Table 3 viruses-17-01196-t003:** Correlations between numerical parameters and IMT at first measurement.

Parameters	Overall Sample (n = 92)			COVID-19 Group			Control Group		
	Rho	95% CI	*p*	Rho	95% CI	*p*	Rho	95% CI	*p*
Age	0.53	0.36–0.67	**<0.001**	0.49	0.22–0.68	**<0.001**	0.58	0.33–0.75	**<0.001**
BMI	−0.11	−0.31–0.11	0.314	−0.14	−0.42–0.16	0.351	−0.06	−0.36–0.25	0.716
Time between first and second measurement	−0.12	−0.39–0.17	0.395	0.07	−0.34–0.45	0.739	−0.43	−0.71–−0.03	**0.031**
IMT at second measurement	0.73	0.56–0.84	**<0.001**	0.51	0.16–0.76	**0.006**	0.87	0.72–0.94	**<0.001**
Change in IMT between measurements	0.06	−0.23–0.34	0.675	−0.09	−0.47–0.32	0.678	0.13	−0.29–0.51	0.531
SBP	0.12	−0.09–0.32	0.259	0.27	−0.03–0.52	0.070	−0.05	−0.34–0.26	0.760
DBP	−0.09	−0.29–0.13	0.409	0.01	−0.28–0.31	0.937	−0.17	−0.45–0.14	0.268
HR	−0.14	−0.34–0.07	0.186	−0.18	−0.45–0.13	0.238	−0.15	−0.43–0.16	0.337
CRP	−0.08	−0.29–0.13	0.453	0.01	−0.29–0.31	0.933	−0.20	−0.47–0.11	0.197
Creatinine	0.12	−0.09–0.33	0.245	0.12	−0.18–0.40	0.414	0.15	−0.16–0.43	0.336
Total cholesterol	−0.09	−0.30–0.13	0.421	−0.30	−0.55–0.00	**0.044**	0.09	−0.22–0.39	0.544
LDL	−0.08	−0.29–0.13	0.441	−0.31	−0.56–−0.01	**0.039**	0.11	−0.21–0.40	0.494
HDL	−0.20	−0.39–0.02	0.066	−0.33	−0.57–−0.03	**0.029**	−0.14	−0.43–0.17	0.363
TG	0.20	−0.01–0.40	0.059	0.29	−0.02–0.54	0.055	0.11	−0.20–0.40	0.470
TG/HDL	0.22	0.01–0.41	**0.038**	0.34	0.04–0.58	**0.022**	0.12	−0.20–0.41	0.453
Chol/HDL	0.14	−0.07–0.35	0.177	0.06	−0.24–0.36	0.687	0.21	−0.10–0.48	0.178
LDL/HDL	0.08	−0.13–0.29	0.444	−0.03	−0.33–0.27	0.827	0.16	-0.15–0.45	0.291

Note: Bold—statistically significant, BMI—Body Mass Index, IMT—intima–media thickness, SBP—Systolic Blood Pressure, DBP—Diastolic Blood Pressure, HR—heart rate, CRP—C-Reactive Protein, LDL—Low-Density Lipoprotein Cholesterol, HDL—High-Density Lipoprotein Cholesterol, TG—Triglycerides, and Chol—total cholesterol.

**Table 4 viruses-17-01196-t004:** Correlations between numerical variables and inter-measurement change in IMT (Δt2 − t1).

Parameters	Overall Sample (n = 92)				COVID-19 Group				Control Group			
	*N_pair_*	Rho	95%CI	*p*	*N_pair_*	Rho	95%CI	*p*	*N_pair_*	Rho	95%CI	*p*
Age	51	0.14	−0.15–0.14	0.316	26	0.07	−0.34–0.45	0.750	25	0.19	−0.24–0.55	0.373
BMI	51	0.05	−0.24–0.33	0.736	26	−0.14	−0.51–0.27	0.485	25	0.10	−0.32–0.49	0.627
t2–t1	51	−0.15	−0.41–0.14	0.308	26	−0.07	−0.46–0.33	0.721	25	−0.21	−0.57–0.21	0.317
SBP	51	0.01	−0.28–0.29	0.966	26	−0.28	−0.61–0.13	0.160	25	0.34	−0.08–0.66	0.096
DBP	51	0.06	−0.22–0.34	0.656	26	−0.11	−0.49–0.30	0.591	25	0.21	−0.22–0.57	0.318
HR	51	0.35	0.07–0.57	0.013	26	0.39	−0.01–0.68	**0.049**	25	0.14	−0.28–0.52	0.493
CRP	50	0.26	−0.03–0.51	0.066	25	0.09	−0.32–0.48	0.654	25	0.12	−0.30–0.50	0.552
Creatinine	50	−0.10	−0.37–0.19	0.499	26	0.02	−0.38–0.41	0.925	24	0.08	−0.48–0.43	0.699
Total cholesterol	50	0.05	−0.24–0.33	0.733	25	−0.02	−0.43–0.39	0.910	24	0.21	−0.21–0.57	0.309
LDL	50	0.08	−0.21–0.36	0.590	25	0.07	−0.34–0.47	0.723	24	0.20	−0.22–0.56	0.326
HDL	50	−0.32	−0.56–−0.04	**0.022**	25	−0.43	−0.71–−0.03	**0.032**	24	−0.28	−0.62–0.14	0.169
TG	50	0.05	−0.24–0.33	0.737	25	−0.01	−0.41–0.40	0.980	24	0.11	−0.31–0.49	0.596
TG/HDL	50	0.21	−0.08–0.47	0.135	25	0.23	−0.19–0.58	0.259	24	0.20	−0.22–0.56	0.329
Chol/HDL	50	0.28	−0.01–0.52	0.052	25	0.28	−0.14–0.61	0.179	24	0.29	−0.13–0.62	0.160
LDL/HDL	50	0.24	−0.05–0.49	0.097	25	0.24	−0.18–0.59	0.247	24	0.30	−0.12–0.63	0.140

Note: Npair—number of pairs; Rho—Spearman correlation coefficient; 95% CI—confidence interval 95%; and *p*—*p*-value of statistical test. Bold-statistically significant BMI—Body Mass Index, SBP—Systolic Blood Pressure, DBP—Diastolic Blood Pressure, HR—heart rate, CRP—C-Reactive Protein, LDL—Low-Density Lipoprotein Cholesterol, HDL—High-Density Lipoprotein Cholesterol, TG—Triglycerides, and t2 − t1—time between the first and second measurement.

**Table 5 viruses-17-01196-t005:** Changes in IMT between the second and first measurements across demographic and clinical categorical parameters.

Parameter	IMT at First Measurement (mm)								
	Overall Sample (n = 51)			COVID-19 Group (n = 26)			Control Group (n = 25)		
	Group 1 ^a^	Group 2 ^a^	*p* ^b^	Group 1 ^a^	Group 2 ^a^	*p* ^b^	Group 1 ^a^	Group 2 ^a^	*p* ^b^
Sex (female vs. male)	0.07 (0.02, 0.19)	0.13 (0.05, 0.37)	0.289	0.11 (0.06, 0.49)	0.15 (0.08, 0.34)	0.677	0.03 (0.01, 0.09)	0.06 (0.04, 0.37)	0.294
COVID (yes vs. no)	0.13 (0.06, 0.50)	0.05 (0.01, 0.20)	**0.018**	-	-	-	-	-	-
COVID-19 stage (acute vs. non-acute)	-	-	-	0.19 (0.06, 0.69)	0.10 (0.06, 0.13)	0.106	-	-	-
COVID-19 severity (mild vs. moderate/severe)	-	-	-	0.08 (0.05, 0.12)	0.19 (0.09, 0.63)	0.087	-	-	-
Smoking status (yes vs. no ^1^)	0.14 (0.05, 0.29)	0.08 (0.04, 0.22)	0.726	0.46 (0.33, 0.59)	0.11 (0.05, 0.31)	0.194	0.06 (0.03, 0.14)	0.04 (0.01, 0.20)	0.970
DM	0.10 (0.05, 0.55)	0.08 (0.03, 0.21)	0.505	0.14 (0.05, 0.63)	0.13 (0.07, 0.21)	0.978	0.07 (0.04, 0.21)	0.04 (0.01, 0.20)	0.481
HT	0.07 (0.04, 0.29)	0.12 (0.05, 0.21)	0.702	0.10 (0.06, 0.61)	0.19 (0.12, 0.22)	0.590	0.05 (0.01, 0.24)	0.05 (0.03, 0.09)	1.000
Heart failure	0.09 (0.04, 0.40)	0.08 (0.04, 0.20)	0.643	0.10 (0.07, 0.50)	0.14 (0.05, 0.31)	0.937	0.05 (0.02, 0.35)	0.05 (0.01, 0.10)	0.579
Vascular incident in the past	0.39 (0.05, 0.78)	0.08 (0.03, 0.21)	0.255	0.39 (0.05, 0.78)	0.13 (0.07, 0.21)	0.644	-	-	-

Note: Bold—statistically significant, ^a^ Median (IQR), ^b^ Wilcoxon rank-sum test. BMI—Body Mass Index, DM2—Diabetes Mellitus type 2, and HT—hypertension *p*—*p*-value of statistical test. ^1^ Here and below.

**Table 6 viruses-17-01196-t006:** Changes in IMT between the second and first measurements across medications taken.

Medications	Changes in IMT Between Second and First Measurement [mm]								
	Overall Sample (n = 51)			COVID-19 Group (n = 26)			Control Group (n = 25)		
	Group 1 ^a^	Group 2 ^a^	*p* ^b^	Group 1 ^a^	Group 2 ^a^	*p* ^b^	Group 1 ^a^	Group 2 ^a^	*p* ^b^
Statin	0.09 (0.03, 0.38)	0.07 (0.04, 0.20)	0.769	0.10 (0.06, 0.61)	0.13 (0.06, 0.22)	0.758	0.03 (0.01, 0.29)	0.06 (0.02, 0.09)	0.890
ACEI	0.08 (0.02, 0.30)	0.09 (0.04, 0.20)	0.891	0.09 (0.05, 0.61)	0.13 (0.06, 0.22)	0.706	0.04 (0.01, 0.26)	0.05 (0.02, 0.09)	1.000
ARB	0.06 (0.05, 0.10)	0.10 (0.04, 0.28)	0.434	0.06 (0.06, 0.11)	0.15 (0.08, 0.59)	0.397	0.05 (0.00, 0.07)	0.05 (0.01, 0.22)	0.610
Beta-blocker	0.08 (0.05, 0.26)	0.09 (0.01, 0.23)	0.260	0.10 (0.06, 0.65)	0.13 (0.08, 0.21)	0.959	0.06 (0.04, 0.10)	0.04 (0.01, 0.28)	0.705
Calcium channel blocker	0.07 (0.02, 0.43)	0.10 (0.04, 0.22)	0.653	0.12 (0.07, 0.57)	0.13 (0.05, 0.31)	0.927	0.05 (0.01, 0.26)	0.05 (0.01, 0.18)	0.722
Alpha-blocker	0.11 (0.06, 0.54)	0.08 (0.04, 0.22)	0.476	0.41 (0.26, 0.56)	0.13 (0.05, 0.31)	0.470	0.06 (0.04, 0.30)	0.04 (0.01, 0.18)	0.530
Diuretic	0.10 (0.05, 0.51)	0.06 (0.02, 0.20)	0.199	0.16 (0.09, 0.69)	0.11 (0.05, 0.19)	0.247	0,05 (0,30, 0,23)	0,04 (0,01, 0,15)	0,420
Aldosterone antagonist	0.06 (0.03, 0.42)	0.09 (0.04, 0.22)	0.815	0.34 (0.20, 0.47)	0,13 (0,05, 0.3!)	0.885	0.04 (0.02, 0.18)	0.05 (0.01, 0.20)	0.911
Sedatives/Hypnotics	0.09 (0.05, 0.40)	0.08 (0.04, 0.21)	0.408	0.14 (0.08, 0.68)	0.12 (0.05, 0.21)	0.331	0.04 (0.02, 0.17)	0.05 (0.01, 0.18)	0.832
Anticoagulants	0.06 (0.05, 0.26)	0.09 (0.03, 0.25)	0.850	0.11 (0.05, 0.39)	0.13 (0.07, 0.41)	0.729	0.06 (0.02, 0.07)	0.04 (0.01, 0.22)	0.759
Hypoglycemic agents	0.08 (0.05, 0.15)	0.08 (0.04, 0.28)	0.776	0.08 (0.05, 0.15)	0.14 (0.08, 0.60)	0.377	0.07 (0.04, 0.18)	0.04 (0.01, 0.20)	0.578
Acetylsalicylic Acid	0.20 (0.05, 0.43)	0.07 (0.04, 0.19)	0.408	0.40 (0.04, 0.78)	0.13 (0.06, 0.21)	0.696	0.20 (0.05, 0.33)	0.04 (0.01, 0.09)	0.324

Note: ^a^ Median (IQR), ^b^ Wilcoxon rank-sum test, ACEI—Angiotensin-Converting Enzyme Inhibitors, and ARB—Angiotensin II Receptor Blockers.

**Table 7 viruses-17-01196-t007:** Adjusted effects of predictors on changes in IMT values in the overall cohort, COVID-19 group, and control group.

Predictor	Overall Sample (*n_obs_* = 51)			COVID-19 Group (*n_obs_ =* 26)			Control Group (*n_obs_* = 25)		
	β	95% CI	*p*	β	95% CI	*p*	β	95% CI	*p*
BMI, kg/m^2^	−1.40 × 10^−3^	−0.01–0.01	0.806	−0.01	−0.03–0.01	0.317	2.67 × 10^−3^	−0.02–0.02	0.770
COVID-19 *[yes, with non-occurrence as ref.]*	0.06	−0.03–0.15	0.216	-	-	-	-	-	-
COVID-19 stage during first measurement *[acute with non-acute as ref.]*	-	-	-	−0.27	−0.54–1.30 × 10^−3^	**0.049**	-	-	-
COVID-19 severity *[mod./severe, with mild as ref.]*	-	-	-	0.24	−0.04–0.51	0.092	-	-	-
IMT at first measurement, mm	0.05	−0.22–0.31	0.733	−0.14	−1.25–0.97	0.800	0.09	−0.18–0.36	0.501
Days between first and second measurement	−2.94 × 10^−4^	−1.1 × 10^−3^–0.50 × 10^−3^	0.438	−5.16 × 10^−4^	−2.0 × 10^−3^–0.90 × 10^−3^	0.464	−1.54 × 10^−5^	−1.8 × 10^−3^–1.70 × 10^−3^	0.986
SBP, mmHg	−4.74 × 10^−4^	−4.00 × 10^−3^–3.00 × 10^−3^	0.786	−0.01	−0.01–0.00	0.234	2.09 × 10^−3^	−1.9 × 10^−3^–0.61 × 10^−3^	0.293
DBP, mmHg	1.40 × 10^−3^	−3.90 × 10^−3^–6.70 × 10^−3^	0.599	−2.80 × 10^−3^	−0.02–0.01	0.724	4.11 × 10^−3^	−1.7 × 10^−3^–9.90 × 10^−3^	0.157
HR	5.90 × 10^−3^	2.50 × 10^−3^–9.30 × 10^−3^	**0.001**	0.01	0.07 × 10^−3^–0.01	**0.033**	3.51 × 10^−3^	−2.1 × 10^−3^–9.10 × 10^−3^	0.205
Smoking status *[yes, with no occurrence intake as ref.]*	0.08	−0.08–0.24	0.335	0.29	−0.12–0.69	0.152	0.06	−0.12–0.25	0.486
DM2 *[yes, with no occurrence intake as ref.]*	0.04	−0.08–0.17	0.486	0.07	−0.26–0.40	0.681	0.04	−0.17–0.24	0.712
HT *[yes, with no occurrence intake as ref.]*	−0.03	−0.16–0.09	0.593	−0.02	−0.26–0.22	0.842	−0.02	−0.20–0.16	0.806
Heart failure *[yes, with no occurrence intake as ref.]*	1.89 × 10^−3^	−0.12–0.12	0.975	−0.01	−0.22–0.20	0.904	−0.03	−0.22–0.15	0.708
Vascular incident in the past *[yes, with no as ref.]*	0.28	0.07–0.49	**0.011**	0.24	−0.10–0.58		**-**	**-**	**-**
CRP	4.26 × 10^−4^	−0.60 × 10^−3^–1.50 × 10^−3^	0.427	1.51 × 10^−3^	−0.8 × 10^−3^–3.80 × 10^−3^	0.191	7.41 × 10^−6^	−2.1 × 10^−3^–2.10 × 10^−3^	0.994
Creatinine	−0.13	−0.32–0.07	0.193	0.23	−0.46–0.92	0.494	−0.17	−0.42–0.09	0.184
Total cholesterol	6.81 × 10^−5^	−1.1 × 10^−3^–1.20 × 10^−3^	0.907	1.51 × 10^−3^	−2.5 × 10^−3^–1.50 × 10^−3^	0.625	5.58 × 10^−4^	−1.5 × 10^−3^–2.60 × 10^−3^	0.570
LDL	3.05 × 10^−4^	−0.90 × 10^−3^–1.60 × 10^−3^	0.625	−1.84 × 10^−4^	−2.1 × 10^−3^–1.70 × 10^−3^	0.836	1.02 × 10^−3^	−1.00 × 10^−3^–3.10 × 10^−3^	0.306
HDL	−2.01 × 10^−3^	−5.10 × 10^−3^–1.10 × 10^−3^	0.198	−3.81 × 10^−4^	−9.4 × 10^−3^–1.80 × 10^−3^	0.174	−5.35 × 10^−4^	−6.00 × 10^−3^–4.90 × 10^−3^	0.839
TG	−1.58 × 10^−5^	−0.90 × 10^−3^–0.90 × 10^−3^	0.971	−1.83 × 10^−4^	−1.0 × 10^−3^–1.40 × 10^−3^	0.761	−2.70 × 10^−4^	−1.70 × 10^−3^–1.20 × 10^−3^	0.700
TG-to-HDL ratio	2.09 × 10^−3^	−0.02–0.03	0.866	4.91 × 10^−3^	−0.03–0.04	0.750	−4.89 × 10^−3^	−0.05–0.04	0.823
Chol.-to-HDL ratio	0.01	−0.02–0.05	0.488	0.01	−0.04–0.06	0.727	0.01	−0.05–0.07	0.685
LDL-to-HDL ratio	0.01	−0.03–0.05	0.547	4.59 × 10^−3^	−0.04–0.05	0.843	0.02	−0.05–0.09	0.559

Note: Bold—statistically significant, β—regression coefficient; 95% CI—confidence interval 95%; and *p*—*p*-value of the statistical test. BMI–Body Mass Index, mod—moderative, IMT—intima–media thickness, SBP—Systolic Blood Pressure, DBP—Diastolic Blood Pressure, HR—heart rate, DM2—Diabetes Mellitus type 2, HT—hypertension, CRP—C-Reactive Protein, LDL—Low-Density Lipoprotein Cholesterol, HDL—High-Density Lipoprotein Cholesterol, TG—Triglycerides, and Chol—total cholesterol.

**Table 8 viruses-17-01196-t008:** Adjusted effects of medications taken on changes in IMT in the overall cohort, COVID-19 group, and control group.

Medications Predictor [Yes, with No Intake as Ref.]	Overall Sample (*n_obs_* = 51)			COVID-19 Group (*n_obs_* = 26)			Control Group (*n_obs_* = 25)		
	β	95% CI	*p*	β	95% CI	*p*	β	95% CI	*p*
ACEI	−0.02	−0.13–0.10	0.737	-	-	-	4.65 × 10^−3^	−0.17–0.18	0.956
Beta-blockers	2.20 × 10^−3^	−0.11–0.11	0.969	0.12	−0.19–0.44	0.431	−0.06	−0.21–0.08	0.361
Calcium channel blockers	−0.01	−0.12–0.10	0.819	3.81 × 10^−3^	−0.24–0.25	0.974	1.55 × 10^−3^	−0.15–0.15	0.983
Alpha blockers	−0.06	−0.12–0.24	0.523	0.24	−0.17–0.65	0.240	−0.06	−0.30–0.18	0.601
Diuretics	0.05	−0.06–0.17	0.324	0.21	−0.09–0.50	0.156	2.82×10^−3^	−0.15–0.16	0.970
Aldosterone antagonists	−0.03	−0.16–0.11	0.674	0.18	−0.28–0.63	0.427	−0.17	−0.37–0.02	0.074
Sedative/Hypnotics	0.04	−0.08–0.15	0.497	0.11	−0.17–0.39	0.424	0.05	−0.12–0.22	0.537
Anticoagulants	−0.09	−0.26–0.09	0.310	−3.31 × 10^−3^	−0.23–0.22	0.976	−0.09	−0.26–0.09	0.310
Hypoglycemic Agents	−0.03	−0.17–0.10	0.630	−0.14	−0.47–0.18	0.371	0.04	−0.17–0.24	0.710
Acetylsalicylic Acid	0.07	−0.06–0.20	0.276	0.27	−0.02–0.55	0.065	0.10	−0.06–0.26	0.205

Note: β—regression coefficient; 95% CI—confidence interval 95%; and *p*—*p*-value of the statistical test. ARB—Angiotensin II Receptor Blockers, ACEI—Angiotensin-Converting Enzyme Inhibitors.

## Data Availability

The original contributions presented in this study are included in the article. Further inquiries can be directed to the corresponding author.

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
