# Peer review of "From Systemic Inflammation to Vascular Remodeling: Investigating Carotid IMT in COVID-19 Survivors"

_viruses, 2025, doi:10.3390/v17091196_

Round 1

Reviewer 1 Report

Comments and Suggestions for Authors

The work by Emilia Bielecka et al., titled "From Systemic Inflammation to Vascular Remodeling: Investigating Carotid IMT in COVID-19 Survivors," is very interesting, but I have some concerns about it.

  • The initial cohort consisted of 92 patients, but follow-up data was only available for 51 (26 in the COVID-19 group and 25 in the control group). This small follow-up sample size significantly limits the statistical power of the longitudinal analysis, making it difficult to draw robust conclusions.
  • Given the high prevalence of asymptomatic infections, it's possible that some individuals in the control group had prior, undiagnosed COVID-19. This could have diluted the observed differences between the groups and underestimated the true effect of the virus on intima-media thickness (IMT) progression.
  • The study does not account for several potential confounding variables that could influence atherosclerosis progression, such as dietary habits, physical activity levels, and socioeconomic status. The authors mention that lifestyle changes during the pandemic could be a confounder, but this is not explored in the analysis.
  • The study performs numerous statistical tests across a wide range of variables, which increases the risk of finding a statistically significant result by chance (Type I error). The authors do not mention any correction for multiple comparisons (e.g., Bonferroni correction), which would make the findings more reliable.
  • In the multiple regression analysis for the COVID-19 group, acute infection at the first measurement was found to be a significant predictor of a smaller change in IMT over time. This is counterintuitive and contradicts the overall trend of accelerated IMT progression in the COVID-19 group. The authors do not offer a clear explanation for this finding, which warrants further investigation and discussion.
  • The authors state that the lack of serological testing is a limitation but then argue that it's not a major issue because of high seroprevalence in the local population. This is a somewhat contradictory argument. High seroprevalence would actually increase the likelihood of asymptomatic infections in the control group, making serological screening even more important for accurate group allocation.
  • The manuscript describes a severity score where "Score 1" indicates patients who required hospitalization and "Score 0" indicates a mild, ambulatory course. However, in Figure 1 and later in the text, the severity is categorized as "mild" and "moderate/severe". This is confusing.
  • The authors should comment on their results in the context of potential new markers, such as Imidazole propionate, e.g.  DOI: 10.1016/j.jinf.2025.106494
Comments on the Quality of English Language

None

Author Response

We sincerely thank the Reviewer for the careful reading of our manuscript and the constructive comments that helped us improve the clarity, methodological transparency, and scientific value of our work. Below, we provide a point-by-point response, indicating the changes made in the revised manuscript. Page and line numbers refer to the revised version with tracked changes.

  1. The initial cohort consisted of 92 patients, but follow-up data was only available for 51 (26 in the COVID-19 group and 25 in the control group). This small follow-up sample size significantly limits the statistical power of the longitudinal analysis, making it difficult to draw robust conclusions.

Response:
We agree with the Reviewer that the reduced sample size in the follow-up phase decreases statistical power and may limit the generalizability of our longitudinal findings. We have now explicitly acknowledged this limitation in the “Limitations” section (page 21, lines 514-516). We believe that 95% confidence intervals for the main longitudinal results to provide a clearer picture of the estimate precision.

The primary limitation of our study is its relatively small sample size, which may have limited the statistical power to detect subtle differences and reduced the generalisability of the findings.”

  1. Given the high prevalence of asymptomatic infections, it's possible that some individuals in the control group had prior, undiagnosed COVID-19. This could have diluted the observed differences between the groups and underestimated the true effect of the virus on intima-media thickness (IMT) progression.

Response:
We thank the Reviewer for this valuable observation and agree that high seroprevalence in the local population increases the likelihood of prior undiagnosed asymptomatic infections among control participants. This misclassification would likely bias the results toward the null, potentially underestimating the true effect of COVID-19 on IMT progression. We have revised the relevant sentence in the Discussion (page 21, lines 516-523) to ensure clarity and avoid the previous ambiguity. The revised text now reads:

“The lack of serological testing (e.g., anti-nucleocapsid antibodies) represents an additional limitation, as it may have led to the inclusion of participants with prior asymptomatic SARS-CoV-2 infection in the control group. Given the high local seroprevalence (>90%), such misclassification is likely [81] However, this would be expected to attenuate, rather than inflate, between-group differences, suggesting that the true effect may be underestimated.”

  1. The study does not account for several potential confounding variables that could influence atherosclerosis progression, such as dietary habits, physical activity levels, and socioeconomic status. The authors mention that lifestyle changes during the pandemic could be a confounder, but this is not explored in the analysis.

Response:

We agree with the Reviewer that the absence of detailed information on dietary habits, physical activity levels, and socioeconomic status is a limitation of our study. While pandemic-related lifestyle changes (e.g., reduced physical activity, altered diet) could potentially confound the interpretation of vascular changes, these variables were not collected and therefore could not be adjusted for in our analysis. Importantly, the study groups did not differ significantly in key baseline characteristics, including BMI and the prevalence of comorbidities, which may suggest broadly similar lifestyle patterns. Nevertheless, we recognize that unmeasured lifestyle differences remain a possible source of residual confounding and have explicitly acknowledged this limitation in the revised manuscript (page 21, lines 528-536). We have also emphasized that future longitudinal studies should incorporate these parameters to allow a more comprehensive assessment of the relationship between COVID-19 and vascular changes.

Moreover, we did not collect detailed information on other important determinants of atherosclerosis progression, such as dietary habits, physical activity levels, and socioeconomic status. While pandemic-related lifestyle changes (e.g., reduced physical activity, altered diet) could have confounded the interpretation of vascular changes, we were unable to adjust for these factors in our analysis. Importantly, the study groups did not differ significantly in key baseline characteristics, including BMI and the prevalence of comorbidities, which suggests broadly similar lifestyle patterns. Nevertheless, unmeasured lifestyle differences remain a potential limitation and should be addressed in future longitudinal studies.” 

  •  

4.The study performs numerous statistical tests across a wide range of variables, which increases the risk of finding a statistically significant result by chance (Type I error). The authors do not mention any correction for multiple comparisons (e.g., Bonferroni correction), which would make the findings more reliable.

Response: We appreciate the Reviewer’s concern regarding the risk of inflated Type I error. In our study, the statistical tests were based on hypotheses specified a priori rather than on an exploratory search across all available variables. For this reason, and to avoid the risk of overly conservative adjustments that could increase Type II error in a relatively small sample, we did not apply a formal multiple-comparison correction. To strengthen the interpretation, we based our conclusions not only on p-values but also on the magnitude of effect sizes, confidence intervals, and consistency of the findings with existing literature. We have now explicitly acknowledged this limitation in the “Discussion” section (page 21, lines 537-540).

“Another limitation of the study is the potential for Type I error due to multiple statistical tests. However, all analyses were hypothesis-driven and defined a priori, and the interpretation of results took into account effect sizes and consistency with prior evidence, rather than relying solely on statistical significance.”

  1. In the multiple regression analysis for the COVID-19 group, acute infection at the first measurement was found to be a significant predictor of a smaller change in IMT over time. This is counterintuitive and contradicts the overall trend of accelerated IMT progression in the COVID-19 group. The authors do not offer a clear explanation for this finding, which warrants further investigation and discussion.

We appreciate the reviewer’s insightful comment regarding the finding that acute infection at the first measurement was associated with a smaller change in IMT over time in the COVID-19 group, which may seem counterintuitive at first glance. One plausible explanation is related to the timing of infection and the subsequent biological response. Patients with a documented acute infection at baseline may have had a relatively shorter follow-up period since infection onset, thus less time for the pathological process leading to intima-media thickening to fully develop. In contrast, those with earlier infections (e.g., several months prior) had more time for vascular remodeling and IMT progression to manifest.

This suggests that the effect of SARS-CoV-2 infection on vascular changes such as IMT thickening may require a latency period before becoming apparent. Therefore, the smaller IMT change in patients with acute infection at the first measurement may reflect the natural time course of vascular injury and repair, rather than contradicting the overall trend of accelerated IMT progression in the COVID-19 group. Further longitudinal studies with more frequent and extended follow-up are needed to elucidate the temporal dynamics of IMT changes post-infection. (Page 19, lines 440-445)

In our regression analysis, acute infection at baseline was linked to a smaller change in IMT over time, which may seem counterintuitive. This could be explained by the shorter time elapsed since infection in these patients, allowing less time for vascular remodeling to occur. The effects of COVID-19 on IMT likely develop over a longer period, so patients infected earlier show greater progression. This temporal aspect should be considered in interpreting the results and warrants further longitudinal study.”

  1. The authors state that the lack of serological testing is a limitation but then argue that it's not a major issue because of high seroprevalence in the local population. This is a somewhat contradictory argument. High seroprevalence would actually increase the likelihood of asymptomatic infections in the control group, making serological screening even more important for accurate group allocation.

We appreciate the reviewer’s insightful comment regarding serological testing and group allocation. We acknowledge that high seroprevalence increases the risk of including asymptomatic infected individuals in the control group, which may dilute observed differences. Our approach prioritized symptomatic status due to logistical constraints and the study design; however, we agree that incorporating serological screening would enhance group classification accuracy. We have clarified this point in the limitations section to better reflect these considerations. (Page 21 528-536)

Moreover, we did not collect detailed information on other important determinants of atherosclerosis progression, such as dietary habits, physical activity levels, and socioeconomic status. While pandemic-related lifestyle changes (e.g., reduced physical activity, altered diet) could have confounded the interpretation of vascular changes, we were unable to adjust for these factors in our analysis. Importantly, the study groups did not differ significantly in key baseline characteristics, including BMI and the prevalence of comorbidities, which suggests broadly similar lifestyle patterns. Nevertheless, unmeasured lifestyle differences remain a potential limitation and should be addressed in future longitudinal studies.” 

  1. The authors should comment on their results in the context of potential new markers, such as Imidazole propionate, e.g.  DOI: 10.1016/j.jinf.2025.106494

Thank you for highlighting the importance of emerging biomarkers such as Imidazole propionate. While our study did not include measurement of this or other novel metabolic markers, we recognize their potential significance in understanding vascular changes and atherosclerosis progression, especially in the context of COVID-19. Imidazole propionate has been implicated in metabolic dysregulation and inflammation, which may contribute to vascular pathology. Future studies incorporating such markers could provide deeper insights into the mechanisms linking COVID-19 with vascular alterations and might improve risk stratification and targeted interventions.(Page 21 Paragraph 409-411)

Nevertheless, future studies incorporating a broader panel of vascular and biochemical markers, including lipoprotein(a) and imidazole propionate, will be essential to more precisely quantify the independent impact of SARS-CoV-2 infection.”

  1. The manuscript describes a severity score where "Score 1" indicates patients who required hospitalization and "Score 0" indicates a mild, ambulatory course. However, in Figure 1 and later in the text, the severity is categorized as "mild" and "moderate/severe". This is confusing.

Thank you for your valuable comment. We have revised the manuscript to clarify the disease severity categorization within the COVID-19 group. Now, "Mild" corresponds explicitly to mild ambulatory cases, and "moderate/severe" refers to moderate/severe cases requiring hospitalization. This consistent classification has been updated throughout the text and in Figure 1 to avoid any confusion.This consistent classification has been updated throughout the text (Page 3 lines 114-119) and in Figure 1 to avoid any confusion.

Reviewer 2 Report

Comments and Suggestions for Authors

The authors aimed to evaluate the potential impact of SARS-CoV-2 infection on the development of atherosclerosis.

The scientific facts are interesting.

The sample size was less. It is good that this fact was admitted as a limitation.

The ratio of tunica media to tunica intima thickness is an early predictor for atherosclerosis. Was such seen? Include this in discussion.

Why does endothelial dysfunction occur? Is vasodilatation impaired? What are the inflammatory events that give rise to such? Please discuss.

The events give rise to cytokine release. What is the role of cytokine storm as it happened during COVID-19 infection? Please discuss such facts.

There is involvement of renin-angiotensin- aldosterone system which gives rise to inflammation, narrowing of blood vessels, proliferation of the muscle in the blood vessel etc. All these facts need to be discussed.

How can the results be influenced with age?

Please discuss the hypercoagulative state which may predispose to atherogenesis.

What is the role of autoimmune system during such infections?

Author Response

We thank the Reviewer for these valuable and constructive comments, which have helped us improve our manuscript. We have addressed each point as follows:

  1. The ratio of tunica media to tunica intima thickness is an early predictor for atherosclerosis. Was such seen? Include this in discussion.
    In line with the Reviewer’s suggestion, we have acknowledged the importance of the tunica media-to-intima thickness ratio as an early predictor of atherosclerosis. While our imaging protocol did not permit separate measurement of these layers, we now highlight its clinical significance and suggest it as a direction for future research.(Added text (Page 22, lines 561-564))

“While our study focused on IMT, the tunica media-to-intima thickness ratio is also an important early predictor of atherosclerosis. Our imaging protocol did not permit separate measurement, but we note its relevance and recommend it for future research. The use of advanced imaging techniques, endothelial function assessments, and genetic profiling could provide additional layers of understanding.”

We appreciate the Reviewer’s recommendation. The Discussion section has been expanded to include a more detailed overview of the possible mechanisms, including endothelial dysfunction, cytokine storm, RAAS involvement and hypercoagulability, as well as autoimmune processes and long-term vascular damage. These additions strengthen the mechanistic framework underlying our findings.

  • Endothelial dysfunction: SARS-CoV-2 infects endothelial cells via ACE2, reducing nitric oxide availability, impairing vasodilation, and increasing vascular stiffness.

Page 189 lines 407-410:

“Studies have shown that COVID-19 may accelerate atherosclerosis through multiple, interconnected mechanisms. SARS-CoV-2 infects endothelial cells via ACE2, reducing nitric oxide availability, impairing vasodilation, and increasing vascular stiffness.”

  • Cytokine storm: The infection triggers a systemic inflammatory response with elevated IL-6, TNF-α, and IL-1β, amplifying vascular injury.

Page 19 lines 410-411:

“The infection triggers a systemic inflammatory response, including elevated IL-6, TNF-α, and IL-1β, which amplifies vascular injury [1].”

  • RAAS involvement and hypercoagulable state: Dysregulation of the renin–angiotensin–aldosterone system promotes vasoconstriction, inflammation, and smooth muscle proliferation, while a prothrombotic state increases the risk of plaque formation and instability.

Page 19 lines 411-414:

“Dysregulation of the renin–angiotensin–aldosterone system further promotes vasoconstriction, inflammation, and smooth muscle proliferation, while a prothrombotic state increases the risk of plaque formation and instability.”

  • Autoimmune system involvement: Persistent vascular damage can remain after viral clearance, driven by low-grade inflammation, extracellular matrix remodeling, endothelial-to-mesenchymal transition, and immune dysregulation, potentially sustained by epigenetic changes.

Page 19 lines 416-418:

“Persistent vascular damage can remain after viral clearance, driven by low-grade inflammation, extracellular matrix remodeling, endothelial-to-mesenchymal transition, and immune dysregulation, potentially sustained by epigenetic changes [53,54].”

  1. How can the results be influenced by age?
    We thank the Reviewer for raising this important point. In line with the suggestion, we have added a note on the modulatory role of age in the vascular effects of SARS-CoV-2 infection. We highlight that older individuals may be more susceptible to IMT changes, consistent with prior evidence linking age with atherosclerosis and vascular remodeling

Page 19, lines 435-438:

Our analysis suggests that age may modulate the extent of IMT changes in the context of SARS-CoV-2 infection, consistent with prior studies showing increased vascular susceptibility in older individuals, because age is a well-known independent risk factor for atherosclerosis and vascular remodeling [63]

Reviewer 3 Report

Comments and Suggestions for Authors

In this cohort study, the authors found a significant difference in carotid IMT between patients with and without a history of COVID-19. The initial hypothesis is quite reasonable and well presented. The manuscript is well organised and sounds scientific. The statistical processing of the data is consistent with the aims and objectives of the study. The results appear interesting and practically useful. However, I would like to make some comments on this study.
1. The difference in carotid IMT between the groups with and without COVID-19 does not fully explain the main role of COVID-19 in accelerating atherosclerosis. The authors could have softened their conclusions on this by adding a discussion of possibly unaccounted risk factors (e.g., Lp(a)), differences in cardiovascular risk, the presentation of comorbid conditions, etc.

2. Although the discussion surrounding the cytokine profile as a result of viral load is well written, I would recommend that the authors add an explanation of the possible molecular mechanisms of further progression of the proliferative response after confirmation of eradication.

Author Response

We sincerely thank the Reviewer for the thoughtful and constructive feedback, which has helped us strengthen the interpretation of our findings and broaden the discussion. In particular, we acknowledge the importance of considering additional cardiovascular risk factors that may have influenced the observed differences in IMT, and we have accordingly refined our conclusions to provide a more balanced perspective. Furthermore, we agree with the Reviewer’s recommendation to expand the discussion of molecular mechanisms that may underlie continued vascular remodeling even after viral clearance, and we have incorporated this into the revised Discussion section.

  1. The difference in carotid IMT between the groups with and without COVID-19 does not fully explain the main role of COVID-19 in accelerating atherosclerosis. The authors could have softened their conclusions on this by adding a discussion of possibly unaccounted risk factors (e.g., Lp(a)), differences in cardiovascular risk, the presentation of comorbid conditions, etc. 

We thank the reviewer for the valuable comments and agree that unmeasured cardiovascular risk factors, such as Lp(a) levels, baseline risk differences, and comorbidities, could contribute to the observed differences in IMT. We have now added this consideration to the Discussion (page 20, paragraph 497-511) to provide a more balanced interpretation of our findings.

While our results indicate a statistically significant difference in carotid IMT progression between individuals with and without prior COVID-19, this finding alone does not fully establish the primary role of SARS-CoV-2 in accelerating atherosclerosis. Unmeasured factors, such as lipoprotein(a) levels or subtle differences in cardiovascular risk and comorbidities, may have influenced the results [82]. However, our multivariable regression analysis (Table 7) partially addresses these concerns by adjusting for several demographic and clinical parameters. In the overall cohort, higher heart rate and a history of vascular incidents were independently associated with greater IMT progression, while in the COVID-19 group, disease severity showed a trend toward a greater increase in IMT and higher heart rate remained a significant predictor. These results indicate that while traditional risk factors remain important, COVID-19 may have an additional, independent effect. Nevertheless, future studies incorporating a broader panel of vascular and biochemical markers, including lipoprotein(a) and imidazole propionate, will be essential to more precisely quantify the independent impact of SARS-CoV-2 infection.

  1. Although the discussion surrounding the cytokine profile as a result of viral load is well written, I would recommend that the authors add an explanation of the possible molecular mechanisms of further progression of the proliferative response after confirmation of eradication.

We also expanded the discussion of molecular mechanisms that may drive continued vascular remodeling after viral clearance. Specifically, we note that persistent low-grade inflammation, endothelial-to-mesenchymal transition, and smooth muscle proliferation—possibly mediated by epigenetic changes—may contribute to IMT progression beyond the acute infection phase. These additions aim to better integrate our results into the broader pathophysiological context. (page 19, 416-425)

Persistent vascular damage can remain after viral clearance, driven by low-grade inflammation, extracellular matrix remodeling, endothelial-to-mesenchymal transition, and immune dysregulation, potentially sustained by epigenetic changes[53,54]. In some patients with long COVID, either PCR-negative or trace PCR-positive viral detection has been reported, with rare cases of persistent PCR-positive shedding beyond 3 months after the initial infection[55]. Such persistence may be associated with an expanded and sustained SARS-CoV-2–specific CD8⁺ T cell response, as observed in Danish post-COVID cohorts, which could further contribute to chronic vascular inflammation and remodeling long after acute infection has resolved. This supports and justifies the validity of our findings[55–57].

Round 2

Reviewer 1 Report

Comments and Suggestions for Authors
  • The study began with 92 participants but only obtained follow-up data for 51 (a 45% loss to follow-up). While the reasons are explained (logistical challenges), the manuscript fails to address this high attrition rate as a potential source of bias in the limitations section. It's possible that the participants who were lost to follow-up were systematically different from those who remained, which could skew the results. This should be discussed by the authors.
  • The finding that acute infection predicted a smaller increase in IMT is counterintuitive. The authors' explanation—that less time had passed for vascular remodeling to occur—is plausible but based on a very small number of patients and requires a more cautious interpretation.
  • The text states that follow-up data was obtained for 26 participants in the COVID-19 group, and this number is used in Figure 4. However, the corresponding chart in Figure 2 reports n=28 for the same group. This discrepancy must be resolved.
  • The median change in IMT for the COVID-19 group is reported as 0.13 mm in the abstract and Table 5. However, the value shown in Figure 4 is 0.12 mm. Similarly, the p-value for this comparison is cited as p=0.018 in the text but p=0.02 in Figure 4. These values should be made consistent throughout the manuscript.
  • The authors should either apply a correction for multiple comparisons (e.g., Bonferroni, FDR) to the correlation tables or provide a stronger justification

Author Response

The study began with 92 participants but only obtained follow-up data for 51 (a 45% loss to follow-up). While the reasons are explained (logistical challenges), the manuscript fails to address this high attrition rate as a potential source of bias in the limitations section. It's possible that the participants who were lost to follow-up were systematically different from those who remained, which could skew the results. This should be discussed by the authors.

We thank the Reviewer for highlighting the issue of attrition. We agree that the relatively high loss to follow-up (45%) is a potential source of bias, as participants who did not return for follow-up may differ systematically from those who completed the study. While the dropout was largely due to logistical challenges, we cannot exclude the possibility that this attrition influenced the findings. We have now acknowledged this in the Discussion section.

Added text: 

One further limitation concerns relatively high loss to follow-up (45%), which raises the possibility of attrition bias. Although the primary reasons for dropout were logistical, participants who were lost may have differed systematically from those who remained, which could have influenced the observed associations. Therefore, the results should be interpreted with caution, and replication in larger cohorts with better follow-up retention is warranted.  ( page: 21, lines 545-550)

The finding that acute infection predicted a smaller increase in IMT is counterintuitive. The authors' explanation—that less time had passed for vascular remodeling to occur—is plausible but based on a very small number of patients and requires a more cautious interpretation.

We appreciate the Reviewer’s comment regarding the finding that acute infection at baseline predicted a smaller change in IMT. We agree that this result should be interpreted with caution, as it is based on a small subgroup and may be influenced by limited statistical power. We have revised the text to highlight this limitation more explicitly.

Added text:

In our regression analysis, acute infection at baseline was associated with a smaller change in IMT over time, which may seem counterintuitive One possible explanation is that, as less time had elapsed since infection, there was less opportunity for vascular remodelling to occur. The effects of COVID-19 on IMT likely develop over a longer period of time, so patients infected earlier show greater progression. However, this interpretation remains tentative and should be viewed with caution, as it is based on a relatively small subgroup and may be influenced by chance variation. Confirmation in larger longitudinal cohorts will be essential. ( page: 19, lines 539-546)

The text states that follow-up data was obtained for 26 participants in the COVID-19 group, and this number is used in Figure 4. However, the corresponding chart in Figure 2 reports n=28 for the same group. This discrepancy must be resolved.

We thank the Reviewer for this careful observation. We have re-checked both the text and the figures, and can confirm that the number of participants with follow-up data in the COVID-19 group is consistently reported as 26 throughout the manuscript (including Figure 2 and Figure 4). We apologize for any confusion.

The median change in IMT for the COVID-19 group is reported as 0.13 mm in the abstract and Table 5. However, the value shown in Figure 4 is 0.12 mm. Similarly, the p-value for this comparison is cited as p=0.018 in the text but p=0.02 in Figure 4. These values should be made consistent throughout the manuscript.

We sincerely thank the Reviewer for pointing out this inconsistency in the reported values (0.018 vs. 0.02 and 0.13 vs 0.12). We apologize for this oversight, which resulted from rounding and formatting during manuscript preparation despite multiple rounds of checking. We have carefully reviewed all statistical values again and corrected the Figure 4 to ensure consistency across the manuscript. (Page 8)

The authors should either apply a correction for multiple comparisons (e.g., Bonferroni, FDR) to the correlation tables or provide a stronger justification

We thank the Reviewer for raising the important issue of multiple comparisons. We agree that conducting numerous statistical tests increases the risk of Type I error. However, given the relatively small sample size and the exploratory aim of identifying potential associations across a wide set of variables, we did not apply a formal correction (e.g., Bonferroni, FDR), as this would result in an overly conservative approach and likely mask potentially relevant findings. Instead, our interpretation does not rely solely on statistical significance but considers effect sizes, confidence intervals, and consistency with previous research. We have now explicitly acknowledged this limitation in the Discussion section.

Materials and Methods added text:

No formal correction for multiple comparisons was applied. This decision is based on the exploratory nature of the study and the relatively small sample size, where conservative adjustments such as Bonferroni or FDR could substantially increase the risk of Type II error and obscure potentially meaningful associations.(page 27, lines 190-193)

Discussion – Limitations added text: 

A drawback of our study is the potential for inflated Type I error due to multiple statistical tests. Because of the exploratory nature of the analyses and the relatively small sample size, we did not apply formal multiple-comparison corrections, which would have substantially reduced statistical power and the ability to detect potential associations. Instead, the interpretation of findings was based on effect sizes, confidence intervals, and their consistency with prior evidence, rather than on p-values alone. It will be necessary for future studies with larger cohorts to confirm these associations under more stringent statistical adjustments. (page 21, lines 537-544 )
